# Persistence Is Multi-Trait: Persistence Scale Development and Persistence Perseveration and Perfectionism Questionnaire into Polish Translation

**DOI:** 10.3390/brainsci13060864

**Published:** 2023-05-26

**Authors:** Wojciech Styk, Szymon Zmorzynski, Marzena Samardakiewicz

**Affiliations:** 1Department of Psychology, Medical University of Lublin, 20-059 Lublin, Poland; 2Department of Cancer Genetics with Cytogenetic Laboratory, Medical University of Lublin, 20-059 Lublin, Poland

**Keywords:** persistence, perseverance, perfectionism, self-control, PPPQ-10, PS-20

## Abstract

Persistence is defined as, among other ways, the need to achieve the goals and strive for the goal. Persistence can also be considered from the perspective of the resource concept, as a positive factor related to an individual’s adaptive behaviour, psychological resilience, and normal self-regulation. In contrast, tendencies behaviourally similar to perseverance, but which may have psychopathological features, are persistence and perfectionism. The main goal of our study was to: (I) Build non-clinical Persistence Scale (PS) in Polish and English; (II) translate in Polish and validate the Persistence, Perfectionism and Perseveration Questionnaire (PPPQ); (III) analyse properties of both scales. Methods: The study was conducted on a non-clinical group of 306 subjects. The mean age was 27.6 and ranged from 18 to 58 years. The properties of both scales were analysed using the NEO-FFI personality inventory, PSS-10 Perceived stress level scale, The UPPS-P Impulsive Behaviour Scale, the SPSRQ Sensitivity to Punishment and Reward scale, Grit scale and NAS-50 Self-Control Scale. Results: The psychometric features of the scales fulfil the requirements for psychometric tools. The factorial structure of both versions of the PS-20 scale proved to be unifactorial. Openness was the only variable to co-occur with the persistence scales of both the PS-20 and the PPPQ-10, and did not co-occur with scales intended to indicate psychopathology (Perseveration, Perfectionism). Negative correlations occurred with variables describing Persistence with levels of perceived stress, anxiety and depressive symptoms. Impulsivity measured by the SUPPS scale also showed negative correlations with the study variables. Conclusions: In the present work, we postulate that persistence is an umbrella construct that gathers and integrates many other traits to form a multi-trait persistence. Perseveration should be regarded as an undesirable trait characterising psychopathological behaviour. Desirable and indicative traits of an individual’s good functioning are persistence and, to some extent, perfectionism. Individuals with low persistence and high perseveration may be characterised by a repertoire of psychopathological behaviours.

## 1. Introduction

In the modern world, people are praised for taking on challenges and persisting through difficulties to achieve the goals set for them or set by themselves. Described trait is called persistence, and is regularly regarded as a necessary element for success [1,2,3,4]. The construct of persistence has yielded a great number of investigations, followed by many conceptualisations and terms or keywords such as the need to achieve the goals and strive for the goal, commitment, self-control, courage, drive, diligence or conscientiousness [3,4,5]. Some studies attempt to conceptualise persistence as a multidimensional construct. The dimension of persistence, for example, can be conceptualised as a construct consisting of persistence in spite of difficulties, persistence in spite of fear and inadequate persistence [6]. Moreover, in some constructs, such as Grit, there is a belief that the difficulty of maintaining effort over a prolonged time is a primary factor influencing persistence [7]. Additionally, the Grit scale is a complex construct composed of two factors, persistence and passion, hence this research points to the relationship of these combined traits collectively referred to as Grit rather than persistence only [8,9]. Persistence is often approached from a motivational perspective, but research also identifies individuals for whom persistence appears to be habitual and is applied to achieve all broad spectrum goals. This concept has led to the notion of a persistent person in whom persistence may manifest as a distinct trait [1,6].

So is persistence an ideal trait? Or does it have a negative side? It seems important to distinguish traits that seemingly could appear to be persistence and nevertheless are not. As Serpell writes in her work, it should be assumed that persistence is voluntary and largely under conscious control. In contrast, acting out of control would be the tendency to continue a particular behaviour or activity beyond the point where it is no longer appropriate or satisfying. Such a tendency to continue an action is not subject to conscious control and is called perseveration [10]. The repertoire of behaviours observed for both persistence and perseveration may be similar, but what makes them different are the cognitive components. A person washing his or her hands for 5 min may be persistent in his or her action or may be perseverative i.e., continuing the action, as he or she tends to continue once started action rather than to discontinue it. These factors have different levels of adaptability and different association patterns with psychopathology [10,11,12]. Recent studies indicate that engagement in maladaptive perseverative behaviour, may be triggered not only by changes in the brain but also by stressors, and can make a healthy brain to be temporarily impaired [12].

The perfectionism cognition theory, proposed in 2016, focused on the cognitive functions of perfectionism, pointing to its relationship with cognitive perseveration [13]. In this model, comparable forms of cognitive perseveration, ruminations and worries play an important role in the perfectionism–distress relationship. The authors noted that perfectionists who are at risk of mental health problems tend to ruminate and worry. Early studies on perfectionism defined as the pursuit of excessively high personal standards and an overly critical evaluation of oneself. The main dimension of this construct is an excessive fear of making mistakes [14]. This original form focused on the maladaptive dimension of perfectionism was also adopted in various eating disorder-related inventories. From this perspective, we can interpret perfectionism as a component related to perseveration [13]. Further studies have extended this theory by proposing multidimensional conceptualisations suggesting that different dimensions of perfectionism affect positive and negative outcomes. Thus, the concepts of Positive and Negative Perfectionism, Maladaptive Perfectionism (MP) and Adaptive Perfectionism (AP), Self-Oriented Perfectionism (SOP), Socially Prescribed Perfectionism (SPP) and Other-Oriented Perfectionism (OOP) were developed [15,16]. Despite these many concepts, perfectionism is strongly associated with eating disorders. Research indicates that maladaptive perfectionism is the one associated with eating disorders, although patients with anorexia nervosa also showed higher levels of adaptive perfectionism [17].

The findings support the hypothesis that perseveration is a risk factor for a wide range of psychiatric disorders, potentially manifesting as compulsive and safety-seeking behaviours. In turn, perfectionism depending on the concept, can be seen as a positive trait, but also as a negative, psychopathological comparable forms of cognitive perseveration [12,13]. It should be noted that perseverative actions, as well as those directed through maladaptive perfectionism can be perceived by other individuals as perseverative, even though they have completely different backgrounds and effects on the person. They also support the hypothesis that persistence is a protective factor, indicating the reward value for effort and potentially explaining the resistance of some individuals in the face of high levels of life stressors [18,19]. Recent studies also indicate that higher grit-perseverance, not grit-passion may be a protective factor in preventing internet addiction [20].

The goal of our work was to build a persistence scale that fits to the concept of persistence as a resource of positive factor related to an individual’s adaptive behaviour, mental resilience and proper self-regulation. We decided to build a new scale because the scales that are currently in use were created for a different purpose and their properties may not fully meet the goal of our study. In addition, recent research on the Grit scale, which is most commonly used in persistence studies, has raised questions about its predictive properties [8,21]. Additional goals of our work were associated with translation and validation of the scale in the English version.

The persistence can also have a psychopathological aspect related to perseverative task performance and psychopathological perfectionism. In order to describe the multi-trait of persistence, we adapted the Persistence, Perfectionism and Perseveration Questionnaire (PPPQ) developed by Lucy Serpell. This questionnaire was constructed as a clinical scale and was designed to detect pathological traits. The scale has mainly been used in studies involving patients with eating disorders [10,11]. We decided to use this scale in our study, because it describes the trait of persistence itself and in addition it describes the pathological dimensions associated with persistence. Despite being a typically clinical scale, it was also validated on healthy individuals. The final goal of our study was to examine the properties of the described scales, by comparing the results with constructs associated with persistence such as personality or conscientiousness.

## 2. Study 1—Construction of the Persistence Scale (PS)

### 2.1. Procedure for Developing Scale Items

In constructing the tool, targeted interviews and dictionary analyses were performed. Next, expert judges were involved in the analysis. In order to build a tool that would accurately describe persistence and be understood by a wide range of different ages people, a questionnaire was first used, in which respondents were asked to complete the sentences “Persistence is...”, “A persistent person is....”, “I am persistent because...”, and to write down associations with the word persistence. In addition, Internet posts on forums, blogs concerning persistence were analysed. In these posts, situations were described (when someone is persistent and when is not), as well as techniques used to increase persistence.

In this way collected data were analysed in order to group them into unified items that will form the tool’s scales. The aim of this stage is to reformulate some terms in such way, that the respondent could refer to the statement on the dedicated scale, but also to ensure that the sentences did not lose their original meaning. The prepared material contained 66 sentences (items) and was subjected to the procedure described below for testing and determining the psychometric properties.

### 2.2. Participants and Procedure

Participants recruited were among full-time and part-time students of several Polish universities. There was no restriction on study field or age. The only exclusions were eating or psychiatric disorders, about which the subjects were asked in the questionnaire.

Study participants were informed about the study’s purpose and that they could withdraw from participation at any time without consequence. After consenting to the study, individuals received a questionnaire containing the original research scale with 66 items. Questionnaires were returned from 96 individuals. The final study included 71 people, of whom 58% were women (*n* = 41) and 42% were men (*n* = 30). The youngest participant was 18 years old and the oldest was 56. The mean age of the individuals was M = 27.61; SD = 2.63. Excluded participants were those who declared eating or psychiatric disorders or did not fully complete the questionnaire.

### 2.3. Results

The analysis strategy consisted of several steps. Initially, it was determined whether the collected data set was sufficient to conduct analyses including factor analysis. The KMO sample adequacy test was used. It had a value of 0.91. The Bartlett’s Sphericity test showed statistical significance (χ^2^ = 1326.58, *p* < 0.001). These results provided the basis for further analysis. In the next step, in order to determine the structure of the tool and to extract the items that will form the scales, an exploratory factor analysis and reliability analysis were conducted. The items so extracted were subjected to stability analysis over time. The last step included a confirmatory factor analysis (CFA) to confirm the obtained results.

The carried out analysis identified a one-factor structure for the tool. The 20 items that loaded the construct most strongly were selected for the final version. The developed univariate structured tool accounted for 62.12% of the total explained variance. Cronbach’s alpha coefficients were used to assess the internal consistency of the scale, which was 0.97. The factor loadings are presented in Table 1.

To determine the stability of the obtained results using the tool, twenty-eight participants in the original study were asked to fill out the scale again (which already contained only the selected 20 items). The results were subjected to correlation analysis, which showed a statistically significant high correlation (r = 0.85; *p* < 0.01) of the obtained results during the one-month interval. Finally, to confirm the results, a confirmatory factor analysis (CFA) was performed. It confirmed a very good fit of the model (RMSEA = 0.05; GFI = 0.98). In order to validate the scale’s properties on a larger sample, re-analyses were performed on data collected in study 4 and including 307 subjects. The conducted analyses confirmed the properties of the scale. The KMO value (0.87) and the result for Bartlett’s sphericity test showed statistical significance (χ^2^ = 1897.95, *p* < 0.001). Factor analysis confirmed the univariate structure. Factor loadings are shown in Table 2. In this study Cronbach’s alpha coefficients obtained the value of 0.93. Confirmatory factor analysis confirmed good fit of the model (RMSEA = 0.06; GFI = 0.93).

The scale was named the Persistence Scale (PS-20). The prefix PL denotes the scale language. The PS-20 scale contains 20 items that form affirmative sentences along with a scale of 1–7 on which the respondent indicates to what extent the sentence applies to him or her. After adding up the results including the inverted items, the final score is calculated. The presented psychometric properties justify the statement that this is a fully validated psychometric tool. The PL-PS-20 scale sheet is attached as Appendix A.

## 3. Study 2—Translation and Validation of the English Version of EN-PS-20

### 3.1. Procedure for Translating the Scale and Participants

The translation procedure involved translation from Polish to English by three researchers fluent in both languages. The results of the three translators’ independent work were compared and analysed to identify discrepancies. The discrepancies were discussed together resulting in modifications or changes in the meanings of some terms. The procedure carried out in this way was intended to improve intelligibility and cultural fit. The prepared translation was then subjected to back-translation (from English to Polish). It was carried out by two other bilingual experts. The back-translated scale was compared with the original version to confirm the accuracy and relevance of the translation. The final translation version was further checked by three native English speakers—experts in the field of psychology. The prepared translation was used to conduct a pilot study in order to confirm the comprehensibility of the sentences. The pilot study was conducted on a group of 11 volunteer nursing students. They did not report that the presented sentences were incomprehensible or that they had difficulty in answering any question.

### 3.2. Participants and Procedure

The study participants were a group of English-speaking students of various disciplines. As in study 1, those with diagnosed eating or psychiatric disorders were excluded from the study as they indicated in the declaration. The study was conducted among 69 students, and for 20 of them English was the national language. The other students were fluent in English. The study was conducted in Poland and the respondents were participants of the ERASMUS program. The age of the individuals was between 19 and 25, with women making up 70% (*n* = 48). Participants were provided with information about the study’s purpose and information about the possibility to withdraw their study consent at any time without consequences. After consenting to the study, participants were given a questionnaire containing a research scale with a request to complete it.

### 3.3. Results

Exploratory factor analysis and confirmatory factor analysis were conducted to validate the English version of the EN-PS-20 scale and to confirm its psychometric properties. KMO’s test of sampling adequacy reached 0.88 and Bartlett’s test of sphericity showed statistical significance (χ^2^ = 1082.91, *p* < 0.001), indicating sampling adequacy and providing a basis for conducting factor analysis. The English version of the PS-20 scale confirmed the one-factor structure of the tool and explained 52.8% of the total variance. Cronbach’s alpha coefficient was 0.92. The factor loadings of each item are shown in Table 2.

Confirmatory factor analysis validated sufficient model fit for the English-language scale (RMSEA = 0.07; GFI = 0.95). The presented psychometric properties of the English-language scale version indicate comparable psychometric properties with the original Polish version. The EN-PS-20 scale is a tool with good psychometric parameters. The EN-PS-20 scale sheet is attached as Appendix B.

## 4. Study 3—Translation and Validation of Persistence Perseveration and Perfectionism Questionnaire

The aim of the present study was to translate into Polish and validate the PPPQ-22 scale developed by Dr. Lucy Serpell. The PPPQ-22 scale is a 22-item self-report questionnaire to measure three constructs, which are associated with a range of psychological disorders. The authors of the scale focused on the psychopathological aspects of the studied dimensions; therefore “Perfectionism” is defined as having high standards for oneself. “Persistence” is defined as the ability to keep going with a behaviour to reach a goal, even when the task is difficult or takes a long time. “Perseveration” is defined as the tendency to continue a particular behaviour, even when it ceases to be effective or rewarding. Reliability and factor analyses confirm these three constructs. Furthermore, they have appropriate test-retest reliability. Clinical group studies using PPPQ-22 suggests that low levels of persistence, rather than high levels of perseveration, may have implications in eating disorders [11].

### 4.1. Scale Translation Procedure and Participants

The scale translation procedure started with an independent translation from English to Polish by three researchers fluent in both Polish and English. In the next step, discrepancies in these translation versions were identified. Once the discrepancies were identified, the meanings of some terms were discussed. Some phrases were modified to improve intelligibility taking into account cultural differences. To confirm the translation accuracy and relevance, the back-translation (from English to Polish) by two other bilingual experts was performed. The back-translated scale was compared by an English-speaking native speaker with the original version. The prepared translation was used to conduct a pilot study for confirming the comprehensibility and relevance of the sentences. The pilot study was conducted on a group of 11 student volunteers and showed that the content of the items was fully comprehensible. The research and analysis were carried out on the primary 28-items. In the original scale, the authors reduced the number of items to 22, discarding 6 items as a result of their analyses. Taking into account the possibility of cultural differences, we decided to carry out research on all 28 primary items. This procedure ensured that we did not miss any item that, due to cultural differences, was not turned on in the original sample and could have been turned on in a sample from other countries.

### 4.2. Participants and Procedure

The study participants were mainly recruited among students from several Polish universities. Both full-time and part-time students were invited to participate in the study. Participants were informed about the study’s purpose, its expected duration and the possibility of withdrawing from the study at any time without consequences. The participants were given a research scale with instructions. Individuals with diagnosed eating or psychiatric disorders were excluded from the study on the basis of their own declaration. A group of 240 subjects was finally enrolled in the study, which consisted of 76% women (*n* = 182) and 24% men (58). The women’s ages ranged from 18 to 58 years with a mean of 28.87; SD = 10.46 and the men’s ranged from 18 to 36 years with a mean of 26.66; SD = 5.87. The mean age of the entire group was M = 27.61; SD = 9.80.

### 4.3. Results

In order to develop, as well as validate the Polish version of the PPPQ scale and confirm its psychometric properties, an exploratory and confirmatory factor analysis was conducted. The KMO sample adequacy test and Bartlett’s Sphericity test were used to determine whether the collected dataset was suitable for conducting a factor analysis. The adequacy test of the collected sample was confirmed by a statistically significant result for the Bartlett’s Sphericity Test (χ^2^ = 2690.26, *p* < 0.001) and the KMO test value (0.78). The analysis confirmed the original structure of the tool, representing Persistence, Perseverance, and Perfectionism. In the validation, there was only a difference in the number of items that were included in the final scale. Taking into account content/item relevance, the analysis identified the 10 items that loaded most strongly on the extracted factors. For the Perseverance subscale, these were items 5, 12, 15 and 20; for Perseverance, items 1, 2 and 6; and for Perfectionism, items 10, 16 and 24 of the 28-item scale. The other excluded items had loadings of less than 0.5 or loaded on more than one factor. All included items were in the final version of the PPPQ-22 scale. Items excluded in the original study, also in the Polish version, were excluded. The three-factor structure accounted for 61.6% of the total variance. Factor 1—Perseverance, explaining 31.61% of the variance, consisted of four items. Factor 2—Perseverance—consisted of three items explaining 12.02% of the variance. Factor 3—Perfectionism—consisted of three items and explained 17.53% of the variance. Cronbach’s alpha coefficients were determined, ranging from 0.62 to 0.79. The results of the analyses are presented in Table 3. Confirmatory factor analysis confirmed sufficient fit of the proposed model (GFI = 0.95; RMSEA = 0.06). The Polish version of the scale consisting of 10 items indicate good psychometric properties, that are comparable to 22-items version. The PPPQ-10 scale questionnaire is attached as Appendix C.

## 5. Study 4—Analysis of the Properties of the PPPQ and PS-20 Scales

### 5.1. Participants and Procedure

In the study, a group of 306 subjects including 232 women (76%) and 74 men (24%) was enrolled. The mean age was 27.6; SD = 9.68 and ranged from 18 to 58 years. It was assumed that this group would be non-clinical therefore people who declared a diagnosed eating or psychiatric disorder were excluded from the study. Participants were mainly recruited among students from several Polish universities. Along with information about study’s purpose, information was provided about the possibility of withdrawing from the study at any time without consequences. After consenting to the study, participants received research questionnaires with instructions and a survey metric.

### 5.2. Methods

#### 5.2.1. Personality

Costa and McCrae’s NEO-FFI personality inventory was used to determine the subjects’ personality traits. This inventory is based on the five-factor model of personality. It is a very popular questionnaire often used not only in research but also in employee training and career counselling, as well as in professional development [22]. The following abbreviations were adopted for the variables described by this inventory: Neuroticism (NEO FFI_N); Extraversion (NEO FFI_E); Openness (NEO FFI_O); Agreeableness (NEO FFI_A); Conscientiousness (NEO FFI_C).

#### 5.2.2. Perceived Stress Level

The PSS-10 scale developed by Sheldon Cohen, Tom Kamarck, Robin Mermelstein was used to assess respondents’ perceived levels of stress. In the study, its Polish adaptation by Juczynski was used which includes ten questions on feelings related to problems and events encountered [23]. The abbreviation PSS-10 was adopted for the variable describing perceived stress level.

#### 5.2.3. Anxiety and Depression

The Hospital Anxiety and Depression Scale (HADS) developed by Zigmond and Snaith was used to measure the levels of anxiety and depression. This screening tool is popular, both in clinical practice and in research. It contains two scales that correspond to levels of general anxiety and depression [24]. The following abbreviations were adopted for the variables described by this inventory: level of depression (HADS_Depression); level of anxiety (HADS_Anxiety).

#### 5.2.4. Impulsiveness

To determine the impulsivity of the subjects, the UPPS-P Impulsive Behaviour Scale was used. The score was determined by three scales. In the study, the Polish version of the scale was used which was developed by Ryszarda Poprawa [25]. The following abbreviations were adopted for the variables described by this inventory: emotion-based rash action (SUPPS_ERA); lack of conscientiousness (SUPPS_LC); sensation seeking (SUPPS_SS).

#### 5.2.5. Sensitivity to Punishment and Reward

The SPSRQ questionnaire was used to measure the Sensitivity to Punishment and Sensitivity to Reward. The Polish adaptation by Wytykowska, Białaszek and Ostaszewski was applied [26]. The following abbreviations were adopted for the variables described by this inventory: sensitivity to punishment (SPSRQ_P); sensitivity to reward (SPSRQ_R).

#### 5.2.6. Self-Control

The NAS-50 Self-Control Scale was used to measure the self-control level. It is a 50-item scale developed by E. Necca containing five subscales [27]. The following abbreviations were adopted for the variables described by this inventory: Initiative and Persistence (NAS_IP); Proactive Control (NAS_PC); Switching and Flexibility (NAS_SF); Inhibition and Adjournment (NAS_IA); Goal Maintenance (NAS_GM).

#### 5.2.7. Grit

To measure Grit, we used the Short version of the scale developed by A. Duckworth and adapted into Polish by Wyszynska et al. Grit consists of Consistency of Interest and Persistence of Effort [28]. In this study was used only the Perseverance of Effort scale (Grit_P).

#### 5.2.8. Data Analytic Procedure

JASP 0.16.3 software was used to perform statistical analyses. The collected data were first analysed to ensure the conditions required for parametric tests [29]. For group comparisons, Student’s *t*-tests and r-Pearson correlation analysis were used taking into account data where parametric tests could be applied. For the remaining data, rho-Sperman analysis and Mann–Whitney test were applied.

### 5.3. Results

First, an analysis was carried out comparing the mean values of the results obtained with regard to the respondents’ gender. Significant differences were found in the mean values of all tested substitutes. Moreover, as regards the group of females higher scores on all scales were observed. The greatest strength of this effect occurred for the PS-20 scale and the lowest for the Persistence subscale of the PPPQ-10 scale. The analysis results are shown in Table 4.

In the next step, the obtained results were analysed with the use of scales describing personality, sensitivity to punishment and rewards, self-control, impulsivity and stress, anxiety and depression. The conducted analyses showed significant, but weak associations of the examined variables with age. Age positively correlates with the studied variables except for the Perseveration, for which the correlation was negative. The correlation analyses with personality indicate that these factors co-occur to varying degrees with the studied variables. The observed correlation of openness with the study variables is noteworthy. This variable is the only one to co-occur only with the persistence scales of both the PS-20 and PPPQ-10, and does not co-occur with the scales intended to indicate psychopathology (Perseveration, Perfectionism). The strongest correlations occurred with conscientiousness, but Perseverance did not correlate with this trait. Correlation analysis with the Perseverance of Effort subscale of the Grit scale showed strong correlations, but only for the PS-20 scale. Sensitivity to punishment appeared to correlate moderately with the PS-20 scale scores and weakly with the PPPQ-10 scale, while sensitivity to rewards showed low correlation only with Perseveration. The components of the self-control scale showed correlations of varying strength with the study variables. The strongest correlations were observed with the scale describing Proactive Control and these correlations were only observed with the PPPQ-10 Persistence and Perfectionism subscales. Negative correlations occurred with variables describing Persistence on both the PS-20 and PPPQ-10 scales with the levels of perceived stress, anxiety and depressive symptoms. Impulsivity, as measured by the SUPPS scale, also showed negative correlations with the study variables, the strongest one with Persistence on both the PS-20 and PPPQ-10 scales. The results of the correlation analyses are shown in Table 5.

## 6. Discussion

The main goal of this work was to build a persistence scale that fits the concept of persistence as a resource, a positive factor associated with an individual’s adaptive behaviours, psychological resilience and proper self-regulation. The constructed scale, after performing the necessary statistical analyses, has emerged a construct with single-factor, which is measured by a 20-item inventory.

For a complete study of the analysed construct, we also decided to adapt the PPPQ scale. The adaptation was needed to examine the psychopathological aspect of persistence associated with perseverative task performance and psychopathological perfectionism and to test the accuracy of the PS scale. The adopted scale contained, similarly to the original version, three factors. However, due to weak factor loadings of original items, only 10 items were finally included in the PPPQ-10 scale.

The components of perfectionism are associated with various negative psychological, interpersonal and physical consequences in people of all age ranges [30]. Birch and colleagues, based on their research, found that self-oriented perfectionism is an adaptive form of perfectionism that promotes flourishing, while socially prescribed perfectionism is a maladaptive form that weakens it. Perfectionism oriented to other individuals is an adaptive form and can enhance well-being [31]. Osenk, on the other hand, found that subscales relating to standards that focus on the pursuit of excellence were positively related to academic performance. The perfectionist concerns were maladaptive to successful learning [32]. In turn, Xie found that both self-oriented perfectionism and socially prescribed perfectionism were positively correlated with worry and ruminations, which contribute to susceptibility to emotional distress and physical disease [33].

Perseveration has been described by Magnin et al. as an abnormal repetition of a motor, behavioural or cognitive process, while Ottaviani characterises it as a cognitive inflexibility that is reflected in both the body and brain [34,35]. Makovac 2020 focuses on perseverative cognition, which is described as intrusive, uncontrolled, repetitive thoughts accompanied by physiological stimulation. Guenther 2020 discusses perseveration of beliefs, which refers to the tendency to maintain beliefs held even when the evidence supporting those beliefs is completely invalidated [36,37]. This work suggests that perseveration includes the persistent repetition of thought, behaviour or belief and may be related to cognitive inflexibility.

As the above works indicate, capturing the dimension of perfectionism and perseverance, despite the fact that it has been functional for many years, is not simple and one-dimensional. The adopted scale describes perfectionist and perseverative tendencies with only three items for each of these variables. It is clear that it is impossible to capture the entire spectrum with these few items. However, we must not forget that the original PPPQ-22 scale was a strictly clinical scale aimed to assess psychopathology. The abbreviated PPPQ-10 scale, in order to fully confirm its properties, should be subjected in the next steps to a study on the relevance of the extracted items and a study on clinical groups. Once its accuracy is confirmed, it could be used as a screening tool to detect psychopathological symptoms associated with, for example, eating disorders, such as the primary scale. So far, the results obtained with this scale should be interpreted with caution.

Regardless of the variables analysed above, it is worth pointing out that in both persistence and self-control there is a conflict of two competing tendencies. One based on the pursuit of a specific goal, the other based on a counter-directed drive pulling us away from it [38]. Goal-oriented behaviour would not be realised without the suppression of competing desires or impulses, an important suppression that often requires great effort [39]. Self-control as studied for many years has often been used as a synonym for other concepts from self-regulation to delayed gratification or self-control strategies such as control of emotions, impulses and desires [38,40,41]. On the basis of action control theory, the term ‘habitual self-control’ is defined as a relatively stable tendency to persistently pursue a goal [42]. The results of our study indicate an association of persistence with various variables describing self-control. Interestingly, depending on the questionnaire, these associations varied in intensity. One of the self-control traits ‘Goal Maintenance’ did not correlate with the persistence dimensions. This trait only correlated negatively with Perseverance and positively with Perfectionism. Goal Maintenance is a variable related to the timeliness of distant plans and meeting deadlines. Perseverance will certainly interfere with meeting deadlines, while perfectionism will be focused not only on the goal but also on the deadline.

Among the Big Five traits, persistence is the most analogous to conscientiousness. Conscientious people are described as accurate, careful, reliable, organised, orderly, efficient, vigilant, but also self-controlled [22]. Conscientious individuals are also dutiful and strive for achievement, these traits differ in that dutifulness is focused on others, while striving for achievement is focused on the self [15]. Consequently, individuals with a high drive for achievement and a low sense of duty may behave in qualitatively different ways from those who are more obedient than achievement-oriented. These subtle but important differences help to separate conscientiousness from persistence. As a trait, persistence should be context agnostic. One who is highly persistent should exhibit this trait regardless of the task, as persistence is only meant to describe a specific behavioural tendency in the pursuit of a goal [6].

The results of our study confirmed that conscientiousness correlated with the persistence dimension. It co-occurred at an average level with both persistence as measured by the PS-20 scale and persistence and perfectionism as measured by the PPPQ-10 scale. Perseverance from the PPPQ-10 scale did not correlate with conscientiousness. As it is a desirable trait that characterises well-functioning individuals [40]. Perseveration was intended to characterise psychopathological behaviour, which only on the surface can resemble conscientious behaviour; this result seems to confirm this assumption giving grounds for treating the perseveration dimension as an undesirable characteristic of psychopathological behaviour. The openness correlated with persistence (measured by both scales). The relationship of openness was not observed with the perseveration and perfectionism. Open-minded people are characterised by a high motivation to seek novelty and unconventional solutions [22]. This motivational factor and the ability to see solutions that can be reached seems to fall also into the broad ‘umbrella’ construct of persistence. Thus, persistent people, in addition to the conscientious approach to tasks that is necessary to achieve a goal, are also characterised by a motivation to act and precisely the openness to change and new experience that is obvious when pursuing a set goal.

Persistence is, in a sense, also the ability to retain (maintain) a previously made choice or chosen goal. This ability to overcome immediate temptations and pursue often distant goals is also observed from the perspective of an individual’s self-control. Lack of self-control, defined as impulsivity, is characterised by the inability to suppress the desire for immediate gratification through a disturbed time horizon. Impulsivity results in a focus primarily on the present and shows very little, if any, interest in the future [43]. These relationships were also confirmed in our study. The negative and strong correlation with the impulsivity subscale—lack of conscientiousness—indicates how impulsivity can decrease the persistence and how it is an important component of this construct.

The ability to postpone immediate gratification depends on many individual and situational factors. Research shows that negative moods and stress lead to poor self-control [38,39,42]. Emotions, especially negative emotions, are one of the factors that influence self-control behaviour [33,44]. The idea that emotional distress interferes with long-term goal attainment and increases the likelihood of succumbing to less favourable short-term goals of immediate gratification has been widely supported by research [45,46]. Anxiety also appears to play a negative role in persistence. Consequently, emotional disturbances such as anxiety and depression make impulsive behaviour more likely [47,48]. In our study, a negative association of both stress and anxiety and depression levels with persistence was confirmed by the PPPQ-10 or PS-20 scales. No association was shown with the perseveration and perfectionism scales. It may indicate that negative effects of these factors on persistence are not related to these variables.

Predicting negative outcomes, in the case of depression or anxiety, can make impulsive behaviour emerge, as individuals become less willing to wait for larger but delayed rewards and choose smaller rewards that are immediately available [49,50]. In contrast, research has shown that individuals with high sensitivity to punishment tend to be less self-controlled than individuals who are less sensitive to punishment [51]. A similar relationship has been shown between neuroticism and sensitivity to punishment [52]. Our results indicate a weak negative association of punishment sensitivity with persistence (stronger for the PS-20 scale), but positive for the Perseveration and Perfectionism scales. Sensitivity to reward was weakly correlated with perseveration. Neuroticism negatively co-occurred with persistence and perfectionism. This confirms the possible negative association of these variables with perseveration and suggests that perfectionism in itself is not necessarily a pathological trait. Possibly, it can become pathological when it is too strong, as other studies have also pointed out [11].

According to research on the original PPPQ-22 scale, both low persistence and high perseveration levels are associated with more unhealthy eating behaviours in this clinical group, in strong contrast to the two comparison groups. In some cases, eating disorder treatment may be enhanced by treating low persistence (and high perseveration). Perfectionism was not elevated in the disorder groups [11]. Our research seems to confirm this trend. Traits that are desirable and indicative of good functioning will be persistence and, to some extent, perfectionism. In contrast, the trait of perseverance will be an undesirable trait. Individuals with low persistence and high perseveration will be characterised by a repertoire of psychopathological behaviours. The findings confirm that perseveration is a risk factor for a wide range of psychiatric disorders, potentially manifesting itself in compulsive behaviours. Persistence is a protective factor, indicating the value of reinforcement for effort and potentially explaining the resistance of some individuals in the face of high levels of life stressors [10,11]. Our research also indicates that persistence is constructed by many traits including self-control, conscientiousness or low impulsivity. Persistence can be described as a multi-trait that integrates many other traits forming the persistent person image. In this model, Persistence is a trait and is habitual. It is used in the achievement of all wide range goals. Recent research on persistence and addiction also suggest that persistence may be a protective trait, e.g., in Internet addictions [20]. These reports indicate how important it is to conduct research on this trait.

## 7. Conclusions

In the present work, we postulate that persistence is an umbrella construct that gathers and integrates many other traits to form a multi-trait persistence. For being persistent, we need both to be conscientious and to have ‘willpower’ for good self-control. The ability to curb impulses and not to put off tasks is essential. The tasks we have to do are not always difficult, they can just be monotonous. However, whatever tasks an individual faces, we see people who are simply persistent regardless of the task type. Perseveration should be regarded as an undesirable trait characterising psychopathological behaviour. Desirable and indicative traits of an individual’s good functioning are persistence and, to some extent, perfectionism. Individuals with low persistence and high perseveration may be characterised by a repertoire of psychopathological behaviours.

### Limitations

Like any study, this one also has its limitations. In terms of adaptation of the PPPQ-10 scale, the main limitation is the inability to examine the accuracy of the constructs of perseverance and perfectionism. In further studies, it would be necessary to compare these dimensions with other scales and determine the relevance of the selected items. In terms of PS-20 scale construction, the main limitations are related to the English version of the scale, which has been adopted into English but has not been culturally adapted to a specific nationality. Despite the use of English, certain phrases may vary from culture to culture. As a result, it is important to make sure that content comprehension is appropriate before starting research in any culture.

The research described in the article was carried out mainly on student groups. Students are a diverse group in terms of age, education level, interests and life experiences. Their participation in the study may contribute to the greater heterogeneity of the research sample, which may affect the representativeness and generalisability of the results. However, students also have specific characteristics such as a particular lifestyle or their involvement in an academic community. Thus, it cannot be ruled out that such group selection may have influenced the results.

## Figures and Tables

**Table 1 brainsci-13-00864-t001:** The factor loadings for PL-PS-20.

Items	Loading *n* = 71	Loading *n* = 306
1. Czas potrzebny na wykonanie zadania nie zmniejsza mojej wytrwałości w osiąganiu celu, nawet gdy jego osiągniecie jest bardzo odległe	0.878	0.647
2. Jestem raczej znany z tego, że potrafię przebrnąć przez przeciwności, aby osiągnąć cel	0.859	0.736
3. Wykonuje zadania tak długo jak potrzeba, nawet gdy potrzeba na to miesięcy czy lat	0.846	0.591
4. Nie mam problemu z wytrwałością, nawet gdy cel jest daleko odsunięty w czasie jak np. ukończenie szkoły czy kursu	0.843	0.621
5. Moi znajomi uznaliby za typowe dla mnie to, że potrafię poświęcić dużo czasu dla osiągnięcia celu	0.835	0.698
6. Jestem wytrwała/y. Nigdy się nie poddaję	0.833	0.731
7. Kończę to, co zaczynam.	0.826	0.706
8. Niepowodzenia nie zniechęcają mnie. Nie poddaję się łatwo	0.817	0.707
9. Jeśli mam problem do rozwiązania, będę długo pracować nad jego rozwiązaniem, niezależnie od tego czy się uda	0.795	0.744
10. Z reguły udaje mi się przezwyciężyć niepowodzenia, aby sprostać ważnemu wyzwaniu	0.787	0.710
11. Czuję wewnętrznie musze skończyć to co zaczęłam/zacząłem	0.784	0.795
12. Kiedy już zdecyduję się coś zrobić, idę dalej, aż osiągam mój cel	0.782	0.693
13. Kiedy postanowię coś zrobić i nie udaje mi się tego dokonać robię wszystko, co możliwe, żeby to zrealizować	0.764	0.698
14. Jestem wytrwały/a ale to tak naprawdę wewnętrzne poczucie sprawia, że kontynuuję jakieś działanie	0.757	0.694
15. Przeważnie potrafię długo być wytrwałą/ym, niezależnie od celu jaki mam osiągnąć	0.753	0.592
16. Podczas nauki do ważnego testu, jeśli zdecydowałam/em się powtórzyć temat robię to aż skończę, nie rozpraszam się	0.741	0.816
17. Mam tendencję do kontynuowania zadania, aż do jego ukończenia, czas nie ma znaczenia liczy się cel	0.720	0.596
18. Czuję się zawsze zobligowany do zakończenia tego to zacząłem	0.715	0.761
19. Przeważnie jestem wytrwały w osiąganiu celów, liczy się zawsze cel, czas nie gra roli	0.710	0.742
20. Często szybko rezygnowałam/em, gdy coś mi się nie udawało *	−0.681	−0.665

* reverse item.

**Table 2 brainsci-13-00864-t002:** The factor loadings for EN-PS-20.

Items	Loading
8. I am persistent. I never give up	0.886
10. I feel internally that I need to finish what I have started.	0.884
9. Failures do not discourage me. I don’t give up easily.	0.864
15. I always feel obligated to finish what I’ve started.	0.864
11. If I decide to do something, I keep going until I reach my goal.	0.860
12. I am persistent, but it is my inner drive that really keeps me going.	0.853
13. When I decide to do something, and I fail—I do everything possible to continue and reach the goal.	0.843
16. I can usually be persistent for a long time, regardless of my goal.	0.831
18. I am usually persistent in achieving my goals; the goal is always the most important for me, the time needed to achieve it doesn’t matter.	0.780
14. I usually manage to overcome failures to overcome an important challenge.	0.767
19. I tend to go about a task until it is completed. Time does not matter but the goal matters.	0.681
17. When studying for an important test, if I have decided to repeat a topic I do so until I finish, without distractions.	0.639
5. When I start a task, I finish it.	0.623
2. I’m rather known to be able to get through adversity to reach my goals.	0.619
4. I perform tasks as long as I need to, even if it can take months or years.	0.602
7. If I have a problem to solve, I will work as long as it’s needed to fix it, even if I cannot come up with a solution.	0.585
20. I often quit quickly when something goes wrong.	−0.551
3. I have no problem with perseverance, even when the goal is far away in time, such as school graduation or an examination	0.512
1. The time needed to complete a task does not reduce my persistence in achieving a goal, even when its achievement is very distant.	0.487
6. My friends would indicate that it is typical for me to spend a lot of time trying to achieve a goal.	0.486

**Table 3 brainsci-13-00864-t003:** The rotated factor solution with Cronbach’s alpha coefficient for PPPQ-10.

Factor Loadings		Items		% Variance/Alfa Cr	
		English	Polish		
Persistence	0.820	I tend to keep going with a long task until it is complete, rather than giving up quickly	Mam zwyczaj kontynuowania długiego zadania aż do jego ukończenia, nie poddaję się szybko	31.61%/0.791	61.6%/ 0.694
	0.753	People describe me as someone who can stick at a task, even when it gets difficult	Ludzie opisują mnie jako osobę, która potrafi wytrwać w zadaniu, nawet jeśli staje się ono trudne.		
	0.648	Once I have decided to do something, I keep going until I reach my goal	Kiedy już postanowię coś zrobić, kontynuuję to, aż osiągnę swój cel		
	0.539	If I try to solve a problem or puzzle, I do not stop until I find an answer	Jeśli próbuję rozwiązać jakiś problem lub zagadkę, nie przestaję, dopóki nie znajdę odpowiedzi.		
Perfectionism	0.530	If I have an important test coming up, I am likely to plan carefully which topics I will need to cover, making a revision timetable to ensure I get everything done	Jeżeli mam przed sobą ważny egzamin, prawdopodobnie dokładnie zaplanuję, jakie tematy będę musiał przerobić i sporządzę harmonogram powtórek, aby mieć pewność, że wszystko zrobię.	17.53%/0.639	
	0.574	When calling a tradesman to arrange for him to come to my home, I would make sure I had all the relevant paperwork and measurements ready	Kiedy dzwonię do handlowca, żeby umówić się z nim na wizytę w moim domu, upewniam się, że mam przygotowane wszystkie odpowiednie dokumenty i pomiary.		
	0.551	If I have an appointment, I always check my travel arrangements carefully in advance to make sure that I have plenty of time to get there and not be late	Jeśli jestem umówiony na spotkanie, zawsze wcześniej dokładnie sprawdzam, czy mam wystarczająco dużo czasu, aby dojechać na miejsce i nie spóźnić się.		
Perseveration	0.793	When I phone someone to get a decision, if I get an engaged tone then I tend to keep ringing back every minute or so, even when the deadline for the decision has passed	Kiedy dzwonię do kogoś, aby uzyskać informację i nie mogę się dodzwonić, to mam zwyczaj dzwonienia co minutę, nawet jeśli informacja nie jest mi już potrzebna.	12.02%/0.618	
	0.654	Sometimes I find myself continuing to do something even when there is no point in carrying on	Czasami przekonuję się, że kontynuuję coś, nawet jeśli nie ma to już sensu.		
	0.559	When calling a tradesman to arrange for him to come to my home, I may continually ring and leave messages on the same number, even though I know that they are not being picked up or responded to	Kiedy dzwonię do handlowca, żeby umówić się z nim na wizytę w moim domu, mogę ciągle dzwonić i zostawiać wiadomości pod tym samym numerem, mimo że wiem, że nie są one odbierane ani nikt nie odpowiada na nie.		

Reprinted/adapted with permission from Ref. [10].

**Table 4 brainsci-13-00864-t004:** Analysis of mean scale values according to gender.

	Group	N	Mean	SD	*p*	Effect Size
PS-20	Female	232	94.28	22.31	<0.01	0.58
	Male	74	106.32	15.13		
PPPQ_Persistence	Female	182	30.02	5.66	0.03	0.34
	Male	58	31.83	4.10		
PPPQ_Perseveration	Female	182	22.22	5.19	<0.01	0.41
	Male	58	24.17	2.90		
PPPQ_Perfectionism	Female	182	22.08	4.56	<0.01	0.44
	Male	58	20.21	2.97		

**Table 5 brainsci-13-00864-t005:** Correlation of PS-20 and PPPQ with the study variables.

	PL-PS-20	PPPQ-10_Persistence	PPPQ-10_Perseveration	PPPQ-10_Perfectionism
AGE	0.21 ***	0.24 ***	−0.24 ***	0.15 *
NEO FFI_N	−0.41 ***	−0.26 **	0.08	−0.21 *
NEO FFI_E	0.28 ***	0.25 **	0.03	0.08
NEO FFI_O	0.20 *	0.25 **	0.20	0.20
NEO FFI_A	−0.05	0.01	−0.05	0.01
NEO FFI_C	0.47 ***	0.40 ***	0.12	0.44 ***
Grit-P	0.61 ***	−0.03	0.01	0.01
SPSRQ_P	−0.37 ***	−0.19 *	0.23 **	0.21 *
SPSRQ_R	0.01	0.03	0.17 *	−0.07
NAS_IP	0.39 ***	0.28 ***	0.20 *	0.32 ***
NAS_PC	0.11	0.43 ***	0.09	0.45 ***
NAS_SF	0.24 **	0.31 ***	−0.03	0.08
NAS_IA	0.25 **	0.14	−0.06	−0.11
NAS_GM	0.01	0.13	−0.34 ***	0.24 **
PSS-10	−0.33 ***	−0.26 **	−0.07	0.18
HADS_Depression	−0.28 ***	−0.41 ***	−0.06	0.03
HADS_Anxiety	−0.23 ***	−0.17 *	0.05	0.12
SUPPS_ERA	−0.29 ***	−0.13	0.11	0.16
SUPPS_SS	−0.16	−0.17 *	0.18 *	−0.10
SUPPS_LC	−0.61 ***	−0.40 ***	−0.06	−0.23 **
PPPQ-10_Persistence	0.61 ***	-	0.18 *	0.52 ***
PPPQ-10_Perseveration	0.13 *	0.18 **	-	0.33 ***
PPPQ-10_Perfectionism	0.25 ***	0.52 **	0.33 ***	-

* *p* < 0.05, ** *p* < 0.01, *** *p* < 0.001.

## Data Availability

Data is contained within the article.

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
