# Peer review of "Persistence Is Multi-Trait: Persistence Scale Development and Persistence Perseveration and Perfectionism Questionnaire into Polish Translation"

_brainsci, 2023, doi:10.3390/brainsci13060864_

Round 1

Reviewer 1 Report

First of all, let me thank you for your excellent research. Certainly, it will contribute to the understanding of personality traits, particularly, Persistence, Perseveration and Perfectionism. Nevertheless, it is necessary to make a few comments, the work on which, in my opinion, will help improve the quality of your manuscript.

1.     The number of references from the last five years used in the Introduction and Discussion should be significantly increased.

2.     In Abstract and in Table 5 you mentioned GIRT scale. As I understand, it is a mistake: GRIT is right.

3.     I recommend that you add a subsection on “Statistical analysis”.

4.     Did you provide an Exploratory Factor Analysis? Could you clarify this point?

5.     References in the text should be formatted according to the rules of the journal. 

Author Response

Dear Reviewer

Thank you for the reviews. We agree with most of the comments and have made changes. We hope that the manuscript in its current version will meet the expectations.

First of all, let me thank you for your excellent research. Certainly, it will contribute to the understanding of personality traits, particularly, Persistence, Perseveration and Perfectionism. Nevertheless, it is necessary to make a few comments, the work on which, in my opinion, will help improve the quality of your manuscript.

Thank you for the submitted comments. They will certainly allow us to improve our manuscript and provide a better understanding of the conducted research.

1.     The number of references from the last five years used in the Introduction and Discussion should be significantly increased. 

Thank you for this comment. The entire introduction as well as the discussion has been thoroughly revised. The literature has been updated and the track of thought is more in line with the objectives of the paper.

2.     In Abstract and in Table 5 you mentioned GIRT scale. As I understand, it is a mistake: GRIT is right.

Yes it was a "typo" and it has been corrected. In addition the manuscript has been checked for typos in the text elsewhere.

3.     I recommend that you add a subsection on “Statistical analysis”.

Thank you for your comments. We have improved the section describing materials and methods including statistical analysis.

4.     Did you provide an Exploratory Factor Analysis? Could you clarify this point?

Exploratory factor analysis was conducted. Tables 1, 2, 3, 4 show the values of the loadings that were identified by this analysis.

5.     References in the text should be formatted according to the rules of the journal. 

References have been adapted to the requirements.

Reviewer 2 Report

I would like to say thanks for the opportunity to review this article.

The article presented has a very interesting theme and important to the scientific community.

It presents the construction and validation of two scales to measure persistence, perseverance and perfectionism.

Overall, the article has a scientific and appropriate writing, including all the components of a good scientific research. The title and abstract are related with the content. The title is too long. The keywords are linked to the research, but two are not Mesh words, which can turn the article more hidden in data bases.

Introduction allows the framing of the theme and the research itself. The main goal is appropriate, although there are three studies included in the article, which can be an overloading of new information, but specially it restrains the information each study presents, becoming which one of them very incomplete, especially in methodology explanation.

Methodology is very incomplete and does not allow to understand the steps done to achieve those results. It seems a lot of steps of a good methodology are missing in the studies presented, and some of the guidelines are not respected. The number of participants to validate the scales are very insufficient, there is no presentation of the translation guidelines used, and the steps done, for translation and back-translation, and validity of that process. The study of the scale construction and validation, does not present any methodology guideline used, does not have a construct validation methodology with experts or that allows to understand if the scale was appropriate before it was tested. There is not clear why there were only university students selected, and who this might affect results. There is no inclusion or exclusion criteria presented, or the methodology of sample selection and data collection.

Results of the psychometric tests are adequate and appropriate. But there is no approach to the results found to exclude items in the scale constructed or why did the PPPQ scale selected for translation was the original and not the validated one. There are only data present to justify the inclusion of items, and it’s not possible to understand the psychometric reason for the exclusion of others.

Discussion is done according to the results of the study but is very insufficient in terms of the research done, which comprehends the translations and construction of instruments. There is no analysis of the psychometric properties of the instruments, or the validity of them to analyse the concepts proposed. In terms of content analysis, there were used some other instruments, but the results and analysis done are theoretical, and not related to the content validity of the scales. The results in terms of findings related to the definitions is very interesting.

There are no limitations of the study presented or future suggestions.

References, regardless the pertinence, are old and outdated (more than half of them are more than 10 years old, and only 23% have less than 5 years).

Thank you.

Author Response

Dear Reviewer

Thank you for the reviews. We agree with most of the comments and have made changes. We hope that the manuscript in its current version will meet the expectations.

I would like to say thanks for the opportunity to review this article.
The article presented has a very interesting theme and important to the scientific community.
It presents the construction and validation of two scales to measure persistence, perseverance and perfectionism.Overall, the article has a scientific and appropriate writing, including all the components of a good scientific research. The title and abstract are related with the content. The title is too long. The keywords are linked to the research, but two are not Mesh words, which can turn the article more hidden in data bases.

Thank you for this comment, the title has been shortened and more adapted to the study findings. We have tried to choose the keywords to most accurately reflect the characteristics of the article and the results.

Introduction allows the framing of the theme and the research itself. The main goal is appropriate, although there are three studies included in the article, which can be an overloading of new information, but specially it restrains the information each study presents, becoming which one of them very incomplete, especially in methodology explanation.

Thank you for this comment. The entire introduction as well as the discussion has been thoroughly revised. So that the train of thought was more in line with the objectives of the work. The methodology section has been improved so that the reader can better follow the flow of research.

Methodology is very incomplete and does not allow to understand the steps done to achieve those results. It seems a lot of steps of a good methodology are missing in the studies presented, and some of the guidelines are not respected. The number of participants to validate the scales are very insufficient, there is no presentation of the translation guidelines used, and the steps done, for translation and back-translation, and validity of that process. The study of the scale construction and validation, does not present any methodology guideline used, does not have a construct validation methodology with experts or that allows to understand if the scale was appropriate before it was tested. There is not clear why there were only university students selected, and who this might affect results. There is no inclusion or exclusion criteria presented, or the methodology of sample selection and data collection.

Thank you for this comment, as mentioned above in the paper the methodology was improved. In addition, for the PL-PS-20 questionnaire, additional analyses were performed so as to confirm the properties of the tool.

Results of the psychometric tests are adequate and appropriate. But there is no approach to the results found to exclude items in the scale constructed or why did the PPPQ scale selected for translation was the original and not the validated one. There are only data present to justify the inclusion of items, and it’s not possible to understand the psychometric reason for the exclusion of others.

To validate the PPPQ scale, we used all the original items, due to the fact that in the original study the items were excluded due to their low loading value. Because of cultural differences, we assumed that the excluded items in the original study might have such strong loadings in ours that it would be worth considering their inclusion. As we described in the supplementary methodology, the key to including items in the PPPQ10 scale was loading above 0.5 and only loading for one factor. As a result, items that were not included in the PPPQ10 scale were not included in the PPPQ22 scale. In the discussion, we further discussed the findings regarding the validation of this scale.

Discussion is done according to the results of the study but is very insufficient in terms of the research done, which comprehends the translations and construction of instruments. There is no analysis of the psychometric properties of the instruments, or the validity of them to analyse the concepts proposed. In terms of content analysis, there were used some other instruments, but the results and analysis done are theoretical, and not related to the content validity of the scales. The results in terms of findings related to the definitions is very interesting.

The discussion was extensively revised to reflect the main objectives of the study. Conclusions regarding further work on the PPPQ10 scale were discussed, as well as the results of the PS20 scale. We hope that the discussion conducted in this way will meet the expectations of the reviewers.

There are no limitations of the study presented or future suggestions.

Thank you for this comment. The limitation section has been added in manuscript.

References, regardless the pertinence, are old and outdated (more than half of them are more than 10 years old, and only 23% have less than 5 years).

Both the discussion and the introduction sections have been improved and the number of older refferences has been reduced. I hope that these changes will be satisfactory.

Reviewer 3 Report

Review of the manuscript titled “Self-report tools for clinical and non-clinical measurement of multi-trait persistence. Studies on the validation of the Polish version of the Persistence Perseveration and Perfectionism Questionnaire and the development of the Persistence Scale" (ms id: brainsci-2295704)

Comments to the author
The manuscript (ms) presents four studies that attempt to validate two scales (the Persistence, Perfectionism and Perseveration Questionnaire (PPPQ) and the Persistence Scale (PS)) in Polish and in English. While these measures and the variables measured may be of interest to the readers of Brain Sciences, I have several suggestions that require to be made before I can recommend publication. These suggestions are outlined below. I hope the authors find the observations to be useful and constructive.

1.       The title states clinical and non-clinical yet the sample is non-clinical (e.g., nursing students). I therefore suggest that the title is changed to reflect this information. The abstract also requires works. The abstract would benefit from a rationale, clearer aims and results, and more detail on the implications of the study and future directions. The explanations could also be written more in line with the conventions of academic journals (e.g., subjects; participants) and the conclusion could be more aligned with the aims of the study. Proofreading is required (e.g., GIRT scale).  

2.       The introduction is currently underdeveloped. A significant amount of work is required to provide a sufficient rationale for the study. There requires to be a stronger rationale for validating a scale and developing another scale, which the introduction, in its current form, does not provide. The introduction provides very little rationale for why the studies are required. The introduction and the discussion include several different concepts (e.g., self-control) and discuss samples (e.g., athletes) that aren’t measured/included in the ms, which makes the work difficult to follow. That is, the ms lacks the “golden thread” throughout the paper. I suggest the authors stick to explaining the key variables, both theoretically and empirically. In addition, a number of the arguments/claims in the introduction and discussion do not have support. Each claim should be supported with a reference.

3.       There is an abundance (some may say an overabundance) of measures for perfectionism so why is another measure required? Perfectionism is measured (and therefore conceptualised) in a specific way in the present study, so the introduction needs to reflect this. The introduction provides a definition of perfectionism and theory related to perfectionism. However, these are inconsistent with the definition of perfectionism provided in study 1 section and the items of the measure. I would argue that the items do not capture perfectionism as a trait or as a cognitive process (or perfectionism at all) which is suggested by the introduction. The authors also state that the PPPQ-10 captures the trait, however, the items appear to be more domain specific rather than capturing personality, especially for perfectionism, or it could be argued that the items about tests and appointments seem to reflect conscientiousness or organisation rather than perfectionism. The measure seems to be trying to capture unidimensional perfectionism, yet perfectionism is most frequently conceptualised and measured as multidimensional (see Hewitt et al., 2003; Smith et al., 2022). As such, my main issue pertains to the conceptualisation and measurement of perfectionism. The authors are required to either provide a rationale for the unidimensional nature of the measure or capture the multidimensional nature of perfectionism. However, I would also argue that the measure would be better without the perfectionism dimension.

4.       I am wondering why there was a need for the Persistence Scale (PS)? The authors require to provide a stronger rationale for developing yet another scale. It is noted in 3.1 that the tool is required to accurately describe persistence. Therefore, does that mean that the PPPQ-10 does not accurately describing persistence? Also, the persistence paragraph in the introduction could more effectively provide a rationale for the measure. In its current form, the explanation leads the reader to expect an observational tool, however, that is not the case. As such, the rationale for the measure could be stronger. In parts, the explanation of persistence does not make sense and could be clearer. The scale was tested on 71 participants, whereas a reasonable sample size for a simple CFA model is about N = 150 (Muthén and Muthén, 2002). I suggest the authors re-do the analysis with the full sample and/or provide ratioanale/support for the sample sizes throughout the studies.

5.       With respect to enhancing clarity, there are instances throughout the methods and results where the reader would perhaps benefit from further information so to enhance the replicability of the study. In particular, who were the samples exactly and did this have any bearing on the results? Further, did they value the domain in which they were engaging and what bearing, if any, might this have had? Could the authors also provide some commentary on the preliminary analyses run for the studies? What measures were taken for missing data? More information is required in the procedure components of the studies to allow for replication (e.g., ethical approval, recruitment of participants). The participants of study 3 are nursing students; I suggest the authors provide rationale for the sample. The methods could also be written more in line with academic language (e.g., the age range of participants). The data analyses section would benefit from more rationale for the steps taken. I suggest the authors also provide full information on model fit (e.g., RMSEA, CFI, TLI, Chi Square) for CFA. The tables require notes to outline the abbreviations used and the explanations of the tables (e.g., Table 5) could be more effective.

6.       In its present form, the paper does not make a strong theoretical contribution to the literature. There must be some consideration of theoretical contribution (or at least broadening the introduction and discussion section around theoretical and practical significance) would improve the paper. Despite the paper providing “validation of scales”, the introduction, discussion, and conclusion do not reflect the aims and therefore would benefit from a clearer link between the aims of the studies and these components of the ms.

7.       The discussion requires works. Similar to the other sections, the discussion could be written more in line with the conventions of academic journals by starting with clear aims and hypotheses and outlining the results in line with the aims and hypotheses. There also has to be a clearer alignment of support/research between the introduction and the discussion. Many of the claims in the discussion are not support with literature. I suggest the authors work hard to back up all claims with references. I also suggest that terminology is consistent throughout the ms. The discussion introduces “strong emotions” for depression and anxiety – I would encourage the authors to use the names of the variables that are examined in the study. The discussion provides information on eating disorders, however, eating disorders were not measured. Some of the explanations could be more effective; for example, “persistence is a protective factor”. I am left asking myself, a protective factor of what? The introduction and the discussion require clearer explanations throughout.

8.       As noted above, the conclusion also requires work. The conclusion could be more in line with the aims of the study and could more effectively summarise the results. The conclusion would also benefit from a strong take home message.

9.       The ms requires proofreading. Provide page numbers for direct quotes or write in your own words to demonstrate understanding.

References

Hewitt, P. L., Flett, G. L., Besser, A., Sherry, S. B., & McGee, B. (2003). Perfectionism Is Multidimensional: a reply to. Behaviour Research and Therapy41(10), 1221-1236.

Muthén, L. K., & Muthén, B. O. (2002). How to use a Monte Carlo study to decide on sample size and determine power. Structural equation modeling9(4), 599-620.

Smith, M. M., Sherry, S. B., Ge, S. Y., Hewitt, P. L., Flett, G. L., & Baggley, D. L. (2022). Multidimensional perfectionism turns 30: A review of known knowns and known unknowns. Canadian Psychology/Psychologie canadienne, 63(1), 16.

Author Response

Dear Reviewer

Thank you for the reviews. We agree with most of the comments and have made changes. We hope that the manuscript in its current version will meet the expectations.

The manuscript (ms) presents four studies that attempt to validate two scales (the Persistence, Perfectionism and Perseveration Questionnaire (PPPQ) and the Persistence Scale (PS)) in Polish and in English. While these measures and the variables measured may be of interest to the readers of Brain Sciences, I have several suggestions that require to be made before I can recommend publication. These suggestions are outlined below. I hope the authors find the observations to be useful and constructive.

Thank you for these critical comments. They certainly raised the quality of the manuscript.

1.       The title states clinical and non-clinical yet the sample is non-clinical (e.g., nursing students). I therefore suggest that the title is changed to reflect this information. The abstract also requires works. The abstract would benefit from a rationale, clearer aims and results, and more detail on the implications of the study and future directions. The explanations could also be written more in line with the conventions of academic journals (e.g., subjects; participants) and the conclusion could be more aligned with the aims of the study. Proofreading is required (e.g., GIRT scale). 

Thank you for this comment. The entire introduction as well as the discussion has been thoroughly revised. The literature has been "refreshed" and the train of thought is more in line with the objectives of the work, which have also been more clearly formulated. We hope that these changes will improve the reception of our work. The work has also been checked for possible typos.

2.       The introduction is currently underdeveloped. A significant amount of work is required to provide a sufficient rationale for the study. There requires to be a stronger rationale for validating a scale and developing another scale, which the introduction, in its current form, does not provide. The introduction provides very little rationale for why the studies are required. The introduction and the discussion include several different concepts (e.g., self-control) and discuss samples (e.g., athletes) that aren’t measured/included in the ms, which makes the work difficult to follow. That is, the ms lacks the “golden thread” throughout the paper. I suggest the authors stick to explaining the key variables, both theoretically and empirically. In addition, a number of the arguments/claims in the introduction and discussion do not have support. Each claim should be supported with a reference.

As above, the introduction has been improved. The information about concepts that are not the subject of this study was removed to make the paper easy to read and understand.

3.       There is an abundance (some may say an overabundance) of measures for perfectionism so why is another measure required? Perfectionism is measured (and therefore conceptualised) in a specific way in the present study, so the introduction needs to reflect this. The introduction provides a definition of perfectionism and theory related to perfectionism. However, these are inconsistent with the definition of perfectionism provided in study 1 section and the items of the measure. I would argue that the items do not capture perfectionism as a trait or as a cognitive process (or perfectionism at all) which is suggested by the introduction. The authors also state that the PPPQ-10 captures the trait, however, the items appear to be more domain specific rather than capturing personality, especially for perfectionism, or it could be argued that the items about tests and appointments seem to reflect conscientiousness or organisation rather than perfectionism. The measure seems to be trying to capture unidimensional perfectionism, yet perfectionism is most frequently conceptualised and measured as multidimensional (see Hewitt et al., 2003; Smith et al., 2022). As such, my main issue pertains to the conceptualisation and measurement of perfectionism. The authors are required to either provide a rationale for the unidimensional nature of the measure or capture the multidimensional nature of perfectionism. However, I would also argue that the measure would be better without the perfectionism dimension.

We thank you for this comment. Of course, there are many measures of Perfectionism and our main goal was not to build another scale. In the revised manuscript, we have indicated that the main goal was to build the PS-20 scale. Validation of the PPPQ scale was "ancillary", so in addition to translating and verifying the structure of the scale, we refrained from substantive analysis of the items. Conclusions and the resulting caution in interpreting the obtained results with this scale are discussed in the discussion section.

4.       I am wondering why there was a need for the Persistence Scale (PS)? The authors require to provide a stronger rationale for developing yet another scale. It is noted in 3.1 that the tool is required to accurately describe persistence. Therefore, does that mean that the PPPQ-10 does not accurately describing persistence? Also, the persistence paragraph in the introduction could more effectively provide a rationale for the measure. In its current form, the explanation leads the reader to expect an observational tool, however, that is not the case. As such, the rationale for the measure could be stronger. In parts, the explanation of persistence does not make sense and could be clearer. The scale was tested on 71 participants, whereas a reasonable sample size for a simple CFA model is about N = 150 (Muthén and Muthén, 2002). I suggest the authors re-do the analysis with the full sample and/or provide ratioanale/support for the sample sizes throughout the studies.

In the rewritten manuscript, we point out the need for a new scale that takes a different look at persistence. We point out the doubts that arise around the GRIT scale. The single-factor structure obtained in our study is different from the scale developed by Howard et.al. (2019). It also indicates that work on this construct should be continued and that not everything has been fully stated yet. To confirm the properties of the tool, we re-analyzed it on the data collected in study 4 on N=306. The data confirmed the properties of the PS-20 scale, and the manuscript was supplemented with the results of this analysis.

5.       With respect to enhancing clarity, there are instances throughout the methods and results where the reader would perhaps benefit from further information so to enhance the replicability of the study. In particular, who were the samples exactly and did this have any bearing on the results? Further, did they value the domain in which they were engaging and what bearing, if any, might this have had? Could the authors also provide some commentary on the preliminary analyses run for the studies? What measures were taken for missing data? More information is required in the procedure components of the studies to allow for replication (e.g., ethical approval, recruitment of participants). The participants of study 3 are nursing students; I suggest the authors provide rationale for the sample. The methods could also be written more in line with academic language (e.g., the age range of participants). The data analyses section would benefit from more rationale for the steps taken. I suggest the authors also provide full information on model fit (e.g., RMSEA, CFI, TLI, Chi Square) for CFA. The tables require notes to outline the abbreviations used and the explanations of the tables (e.g., Table 5) could be more effective.

(1)The description of method procedures has been expanded. Note that English-speaking nursing students appear only in a pilot study to test the comprehensibility of the sentences:  "Prepared translation was used to carry out a pilot study to confirm the comprehensibility and accuracy of the sentences. The pilot study was carried out on a group of 11 volunteer nursing students. "

(2) Various measures of RMSEA and GFI model fit and measures of sampling adequacy are reported in the results.

(3) The methods section includes abbreviations for the scales used. In addition, an index of abbreviations that were used is included at the end of our manuscript. In our opinion, this form will be more readable than the list of abbreviations below the table.

6.       In its present form, the paper does not make a strong theoretical contribution to the literature. There must be some consideration of theoretical contribution (or at least broadening the introduction and discussion section around theoretical and practical significance) would improve the paper. Despite the paper providing “validation of scales”, the introduction, discussion, and conclusion do not reflect the aims and therefore would benefit from a clearer link between the aims of the studies and these components of the ms.

The paper in question is an empirical work focused on the development of effective psychometric measures. The paper already has a significant volume and in our opinion, overextending the introduction is pointless and a detailed description of existing concepts and measures should rather be developed in a review article or meta-analysis. As we indicated earlier, the manuscript has been revised to make the purpose more clear for the reader and the provided information has been selected to give the reader the needed knowledge to understand the steps being taken in our study. We refer the more curious reader to the literature and review papers.

7.       The discussion requires works. Similar to the other sections, the discussion could be written more in line with the conventions of academic journals by starting with clear aims and hypotheses and outlining the results in line with the aims and hypotheses. There also has to be a clearer alignment of support/research between the introduction and the discussion. Many of the claims in the discussion are not support with literature. I suggest the authors work hard to back up all claims with references. I also suggest that terminology is consistent throughout the ms. The discussion introduces “strong emotions” for depression and anxiety – I would encourage the authors to use the names of the variables that are examined in the study. The discussion provides information on eating disorders, however, eating disorders were not measured. Some of the explanations could be more effective; for example, “persistence is a protective factor”. I am left asking myself, a protective factor of what? The introduction and the discussion require clearer explanations throughout.

The discussion section has been improved and supplemented. Terminology was also standardized and threads that were not directly related to the topic of the work were removed. The topic of eating disorders is important to the work insofar as the PPPQ scale was dedicated to their study and this topic cannot be omitted for the full knowledge of the reader.

8.       As noted above, the conclusion also requires work. The conclusion could be more in line with the aims of the study and could more effectively summarise the results. The conclusion would also benefit from a strong take home message.

As indicated above, the discussion has been improved. We hope that in its present form it will be accepted.

9.       The ms requires proofreading. Provide page numbers for direct quotes or write in your own words to demonstrate understanding.

The entire manuscript was checked for any errors, typos or missing footnotes.

Round 2

Reviewer 2 Report

Dear authors,

With the revision, many of the suggestions were considered.

We highlight:

The article has a scientific and appropriate writing, including all the components of a good scientific research. The title, abstract and keywords are related with the content, and were adjusted to the content. There are still some keywords not indexed.

The introduction allows the framing of the theme and the research itself. The main goal is appropriate.

Methodology was revised. There is still an insufficient number of participants in study 1. The methodology of construction, validity and translations were clarified. There is still not clear why there were only university students selected, and who this might affect results. There is no inclusion or exclusion criteria presented, or the methodology of sample selection and data collection.

Results of the psychometric tests are adequate and appropriate, and discussion was improved. Limitations of the study were added and references updated

Thank you.

Author Response

Thank you for your comments.

In study 1 the results were presented on two samples of N=71 and N=306 individuals. Statistics describing the adequacy of the sampling were presented in the text. In addition, as the study [1] indicates, a sample size of at least N= 150, gives a sufficient test power. In our opinion, the statement that the study group is too small is not justified.

Inclusion and exclusion criteria were described in the text. Individuals declaring psychiatric disorders and/or eating disorders were excluded from the study. The sample selection was randomized and therefore it was not necessary to describe it in the article.

Limitations associated participation of students in the study were added to the LIMITATION section:

"The research described in the article was carried out mainly on student groups. Students are a diverse group in terms of age, education level, interests and life experiences. Their participation in the study may contribute to the greater heterogeneity of the research sample, which may affect the representativeness and generalizability of the results. However, students also have specific characteristics such as a particular lifestyle or their involvement in an academic community. Thus, it cannot be ruled out that such group selection may have influenced the results."

We hope that the provided explanations will be satisfactory.

  1. Muthén, L. K., & Muthén, B. O. (2002). How to use a Monte Carlo study to decide on sample size and determine power. Structural equation modeling9(4), 599-620.

Reviewer 3 Report

Dear authors,

Thank you making the revisions to the paper which has improved the work. However, I still have two suggestions to improve the manuscript. 

The introduction still requires to be more focused with a stronger rationale for the study. For example, the perfectionism section still requires work and to be in line with the measurement. That is, the cognitive component of perfectionism is not captured with the scale and therefore the introduction should reflect this. The explanations of perfectionism in the discussion could also mirror the introduction more effectively. 

A stronger rationale and explanations for all samples (e.g., university students) could be provided to enhance the methods sections. 

Author Response

Thank you for your comments.

The goal of the presented work was to build a persistence scale and present its properties (vide the goal described in the introduction section of the manuscript). Perfectionism in itself is not a construct that is the focus of the presented research. In our view, expanding the introduction section on a construct that is not the focus of the study is unnecessary. Previously, Reviewer have suggested the removal of unnecessary references including for example topics associated with sports or eating disorders. In our opinion, expanding the theoretical introduction to include a construct that is not the primary subject of the study should not be expanded for similar reasons.

Limitations associated participation of students in the study were added to the LIMITATION section:

"The research described in the article was carried out mainly on student groups. Students are a diverse group in terms of age, education level, interests and life experiences. Their participation in the study may contribute to the greater heterogeneity of the research sample, which may affect the representativeness and generalizability of the results. However, students also have specific characteristics such as a particular lifestyle or their involvement in an academic community. Thus, it cannot be ruled out that such group selection may have influenced the results."

We hope that the provided explanations will be satisfactory.